

# GEsture: an online hand-drawing tool for gene expression pattern search

Chunyan Wang[1], Yiqing Xu[1], Xuelin Wang[1], Li Zhang[1], Suyun Wei[1], Qiaolin Ye[1], Youxiang Zhu[1], Hengfu Yin[2], Manoj Nainwal[3], Luis Tanon-Reyes[4], Feng Cheng[5], Tongming Yin[6] and Ning Ye[1]

[1] College of Information Science and Technology, Nanjing Forestry University, Nanjing, Jiangsu, China
[2] Key Laboratory of Forest genetics and breeding, Research Institute of Subtropical Forestry, Chinese Academy of Forestry, Hangzhou, Zhejiang, China
[3] Department of Computer Science, Nantong University, Nantong, Jiangsu, China
[4] Department of Cell Biology, Microbiology and Molecular Biology, University of South Florida, Tampa, United States of America
[5] Department of Pharmaceutical Science, College of Pharmacy, University of South Florida, Tampa, United States of America
[6] College of Forest Resources and Environment, Nanjing Forestry University, Nanjing, Jiangsu, China

## ABSTRACT

Gene expression profiling data provide useful information for the investigation of biological function and process. However, identifying a specific expression pattern from extensive time series gene expression data is not an easy task. Clustering, a popular method, is often used to classify similar expression genes, however, genes with a 'desirable' or 'user-defined' pattern cannot be efficiently detected by clustering methods. To address these limitations, we developed an online tool called GEsture. Users can draw, or graph a curve using a mouse instead of inputting abstract parameters of clustering methods. GEsture explores genes showing similar, opposite and time-delay expression patterns with a gene expression curve as input from time series datasets. We presented three examples that illustrate the capacity of GEsture in gene hunting while following users' requirements. GEsture also provides visualization tools (such as expression pattern figure, heat map and correlation network) to display the searching results. The result outputs may provide useful information for researchers to understand the targets, function and biological processes of the involved genes.

## INTRODUCTION

Gene expression profiling (such as Microarray and RNA-seq) data provides important information for researchers to investigate biological function and process. Many public databases including gene expression omnibus (GEO) (*Barrett & Edgar, 2006*), gene signatures database (GeneSigDB) (*Culhane et al., 2012*), and molecular signatures database (MSigDB) (*Liberzon, 2014*) are available to identify the relationship between gene expression and biological functions/processes. For many biological studies, researchers hope to find genes showing "anticipated" expression patterns. For example, biologists know that during a day, the expression levels of light rhythm genes change (increase or

Corresponding author
Ning Ye, yening@njfu.edu.cn

decrease) with the intensity of light, and change back with the darkness of night. However, it is hard for them to find the genes with this particular pattern from large gene expression datasets without a strong bioinformatics background.

Multiple approaches have been developed to find genes showing similar expression patterns across all samples (*Androulakis, Yang & Almon, 2007*; *Sharan & Shamir, 2000*). Of these approaches, clustering is mainly used to solve the problem (*Eisen et al., 1998*; *Jiang, Tang & Zhang, 2004*; *Schliep et al., 2005*; *Wen et al., 1998*). Clustering algorithms include hierarchical clustering (*Jiang, Pei & Zhang, 2003*), self-organizing maps (*Tamayo et al., 1999*), K-means clustering (*Tavazoie et al., 1999*; *Wu, 2008*) and so on. And many clustering approaches indeed performed well in grouping genes with similar expression patterns without any prior knowledge. Balasubramaniyan (*Balasubramaniyan et al., 2005*) proposed the CLARITY algorithm using a local shape-based similarity measurement to dig for similar expression genes. *Qian et al. (2003)* proposed a local clustering algorithm to identify genes with time-delayed and inverted expression patterns ('time-delayed' is defined as gene expression with a time difference, but the overall expression trend is the same, and 'inverted' refers to the fact that genes show high expression levels while some other genes show low expression levels at the same time). Xia designed the eLSA package (*Xia et al., 2013*), which filters out insignificant results and constructs a partial and directed association network.

Unfortunately, the clustering algorithms have some disadvantages. First, the computation complexity of clustering algorithms exponentially increases as the dataset becomes larger. Second, the issue of determining the optimum cluster number is not yet rigorously solved (*Yeung, 2001*). Third, during the data processing, expression data vary greatly. Clustering algorithms generally require pre-processing the original data, and different clustering algorithms will choose different initial partition, which may cause loss of useful information (*Ye et al., 2015*). Fourth, unrelated groups may be merged into one cluster. Fifth, not always can clustering algorithms cluster all the categories, and even a category will be divided into several categories. For example, classical K-means clustering extracts categories of a given number from the gene expression profile. However, it often separates a big similar category into different categories. Because a time-delayed phenomenon often appears in gene expression, K-means clustering cannot recognize it and mistakenly divides it into many categories instead of classifying only one category. Last but not least, they cannot guarantee that they can always find a gene expression pattern that users want to search at any time. In a word, clustering methods can help to understand global profiles of gene expression, but not efficiently enough to detect genes with user-defined expression patterns.

In this paper, we presented an online tool, GEsture, short for Gene Expression gesture. The program searches specific gene expression pattern from time-series gene expression data using an anticipated gene expression pattern drawn by the user instead of using clustering algorithms. GEsture addresses the current shortcomings of the clustering algorithm and allows users to analyze time-series data from a different angle. This method not only can identify co-expression genes but also can detect opposite and time-delayed expression genes. Furthermore, it provides a user-friendly interface for users to input and

visualize the results. The output results may provide useful information for researchers to understand the targets, function and biological processes of the genes of interest.

## MATERIALS & METHODS

### The workflow of GEsture

The primary function of GEsture is to identify genes showing specific patterns from gene expression data. The workflow is illustrated in Fig. 1. The first step is uploading time-series gene expression data. Two modes of operations, user-defined pattern and K-means clustering, are then provided for pattern searching. For user-defined pattern, users can either draw an expression curve by the mouse on the drawing board or select a pre-defined gene expression pattern in the system to search. Classical K-means clustering method extracts expression patterns from the gene expression profile based on the category number assigned by the users.

There are three functions for gene expression pattern searching in GEsture: brush pattern search, contrast pattern search, and shift pattern search.

1. **Brush pattern search (co-expression pattern search).** It is the default pattern search function in GEsture. Users can draw a gene expression curve with mouse on the drawing board. GEsture will identify the genes showing similar patterns (co-expression genes) with the drawn curve. It is noted that users should include as many time points as possible in curve drawing to achieve accurate matching.

2. **Contrast pattern search.** This function searches the genes showing opposite expression pattern to the user's input. It aims at helping users to explore negatively regulated genes. For example, target genes of a transcription factor that inhibits expression can be found using this function.

3. **Shift pattern search.** This function is designed to find genes showing similar but time-delayed (or advanced) expression patterns. It will help users to identify possible downstream/upstream genes. The range of −6 to 6 can be chosen for the shifting gene expression search.

The expression levels of output genes are shown by the heat maps. The network map is generated to display the relationship of the search results.

### Data analysis process

GEsture takes a curve as an input, and it allows the user to search genes with similar expression patterns. As a result, users can see the gene expression curves intuitively rather than in abstract parameters and data. A raw dataset uploaded by the user will be checked to filter low-quality (such as missing and low entropy) data in the uploaded file by GEsture. The search process includes the drawing of an anticipated curve, followed by fitting the system in a line and sampling the data. Afterwards, genes are compared with each other in the gene expression file to calculate the similarity between them. Lastly, an assessment function is adopted using the Pearson correlation coefficient (*Horyu & Hayashi, 2013*; *Wang, Mo & Wang, 2015*) to select closely-related genes. It may take a while when performing this kind of search on a large dataset, but it is significantly faster than clustering. GEsture only compares every gene expression curve in the file, while clustering needs to

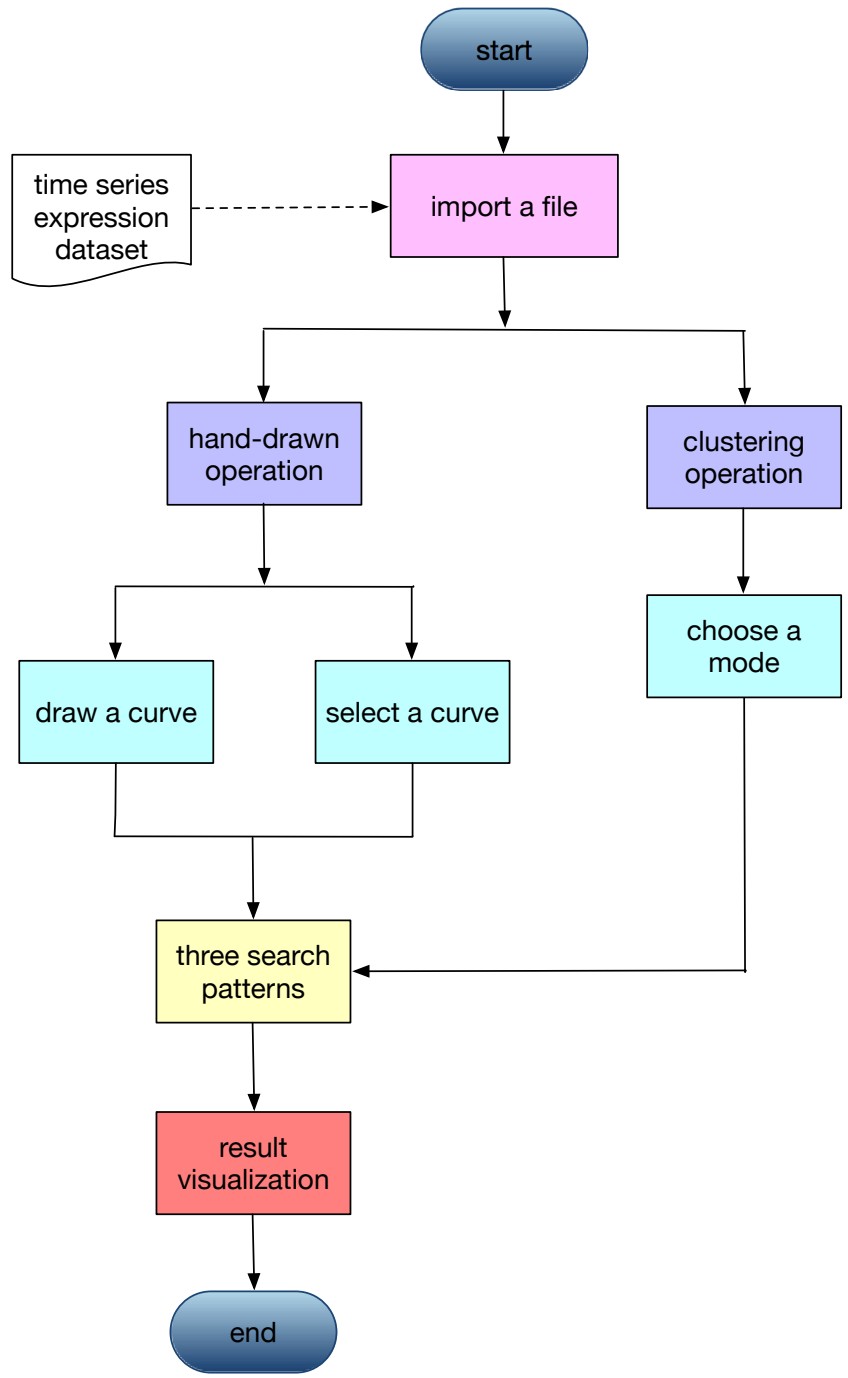

**Figure 1  The workflow of GEsture.** Lists the the main operation procedures of this tool. Rectangular boxes with the same color represent that they are in a parallel relationship.

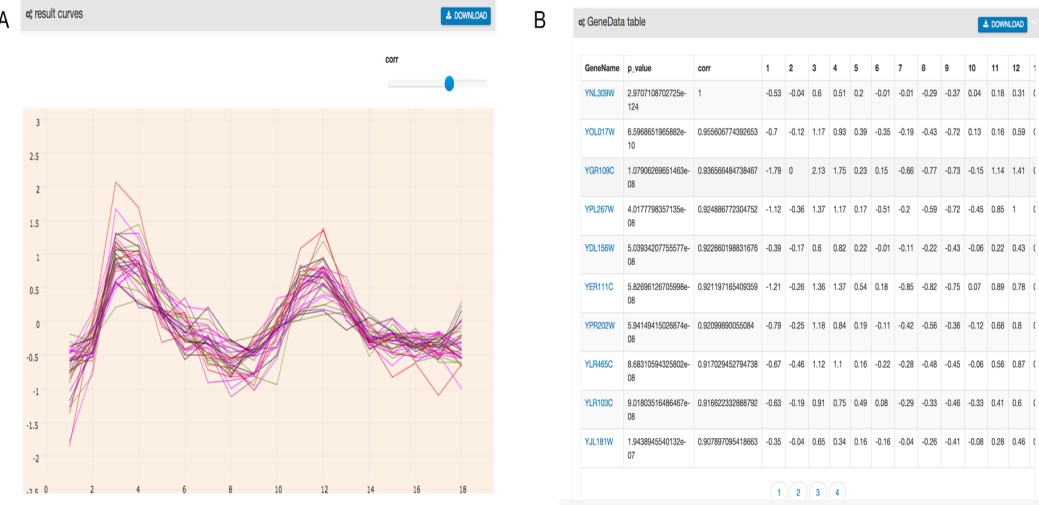

**Figure 2 Filtered result gene curves and the result gene table.** (A) Result gene curves. A slider for users to change the correlation value, enable users to filter data and focus on high correlation genes. (B) The result gene table. With the change of the result curves, table content get changed with it.

determine the initial centers and iteration numbers of the algorithm, inevitably leading to higher computation and time complexity. The cutoff for the correlation coefficient for gene outputs can be adjusted by users if too many or too few genes are identified.

## Output of GEsture

As shown in Fig. 2, the output of GEsture includes a gene expression pattern figure and gene information table. To clearly show the expression patterns, a slider is provided for users to adjust the correlation coefficient cutoff value. At the same time, the information of the output genes in the figure is also shown in a table. In the output table, each row of data represents a gene. The information of gene name, *p*-value, correlation value and detailed time-series expression data is included. If a user clicks on the gene name in the table, the corresponding gene expression curve will be shown in the expression pattern figure. The gene information table can be exported as a CSV file.

Two visualization tools, a heat map and a comprehensive relationship network map were provided to visualize the search results. Figure 3 shows the co-expression genes of YNL309W. Each row in the heat map represents one gene and different colors to display the gene expression levels. In GEsture, the maximum number of genes for a heat map is 500. The heat map can be exported as a PNG formatted file. In addition, a gene network is used for representing the complex functions or traits of biological system, especially the network based on co-expression genes can annotate the unknown gene function (*Serin et al., 2016*). But here, we build a simple 'network' to show the output genes who may have the latent biological relationship searched by three searching patterns, which is shown as Fig. 4. In Fig. 4, circles in the same color represent they are similar expression genes and the center point is the searched gene or the gene most similar to the drawn-curve. It was built on the Cytoscape.js, and the size of Fig. 4 can be adjusted by users using the mouse.

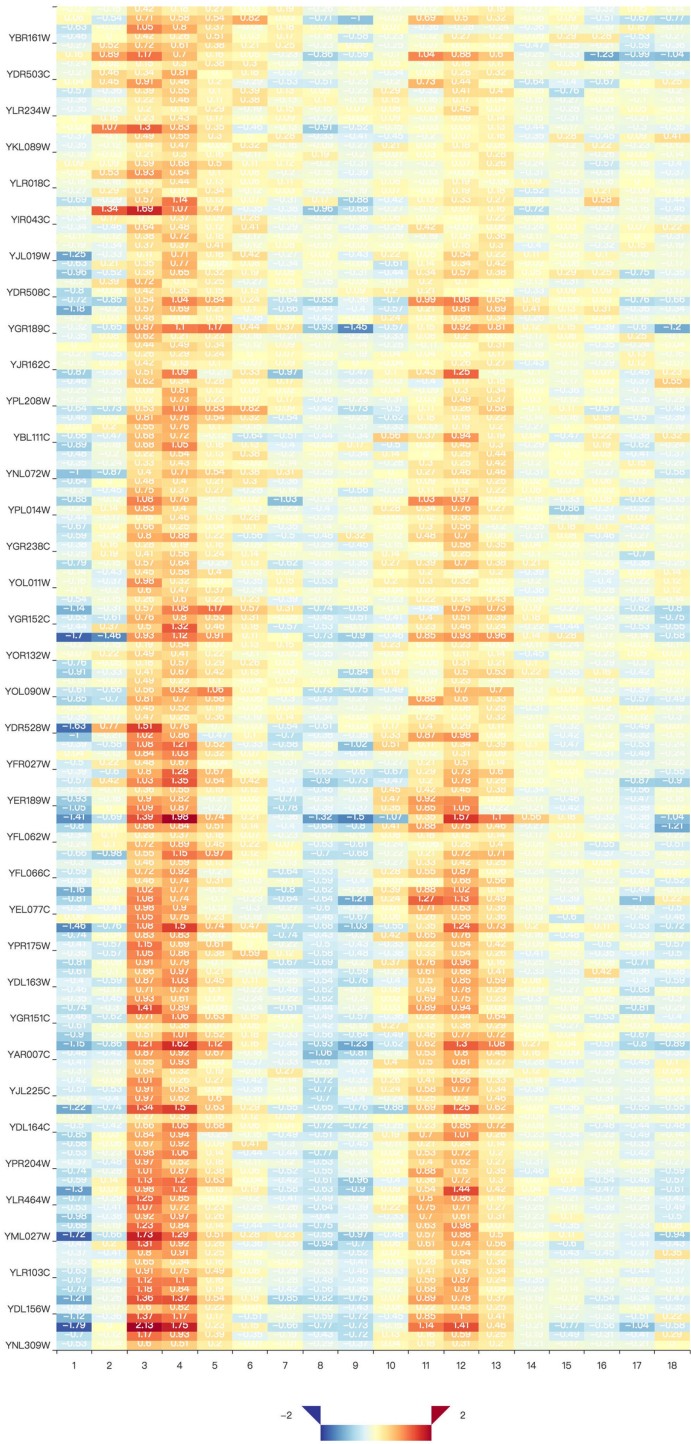

**Figure 3  The heat map of co-expression genes of YNL309W.** The darker the red color, the higher the gene expression level and the darker the blue, the lower the gene expression level.

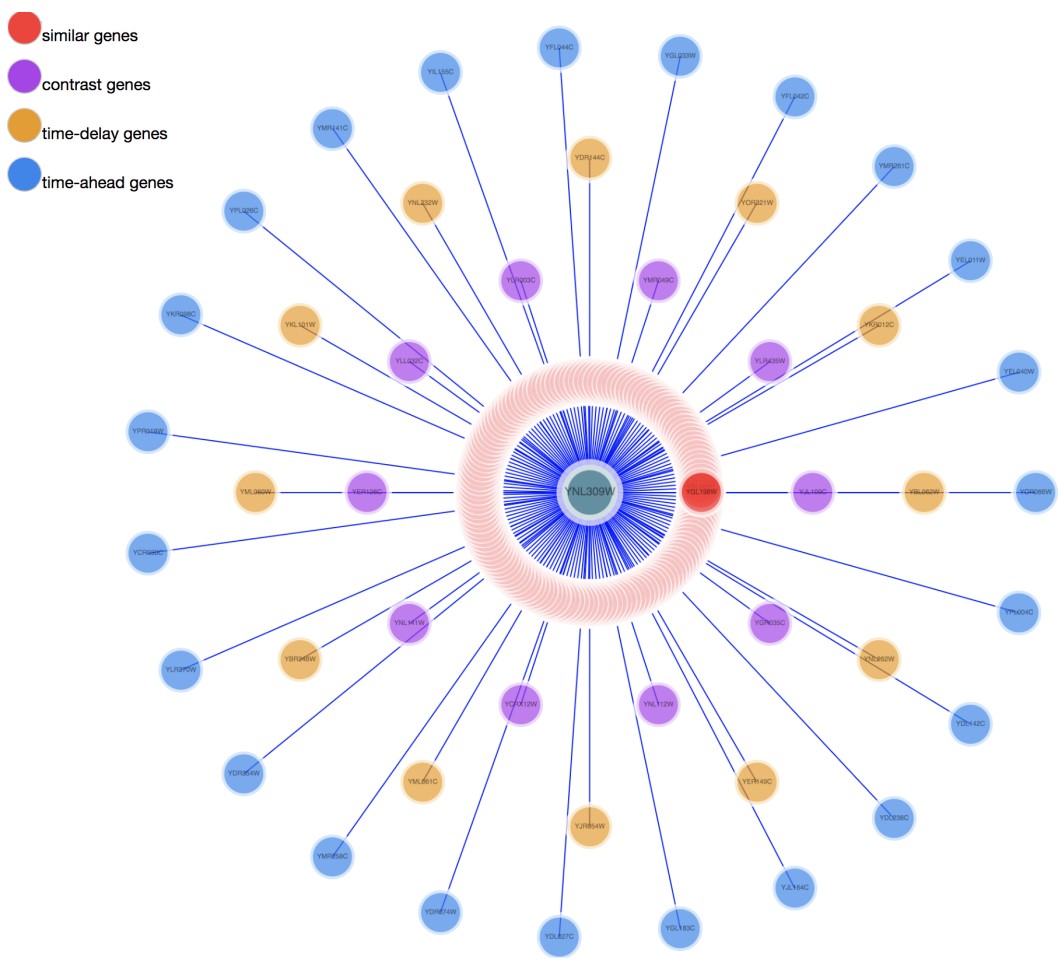

**Figure 4   The network map of YNL309W.** The red circle are genes searched out by similar pattern. The purple circles represent opposite expression genes. The yellow circles are time-delay expression genes and the blue circles are time-ahead expression genes.

## Datasets used in GEsture

We used three examples to demonstrate the effectiveness of GEsture. In example 1, a yeast cell-cycle dataset was chosen to assess the performance of GEsture. The dataset contains 6187 genes and 18-time points (*Spellman et al., 1998*), and it is available at http://genome-www.stanford.edu/cellcycle/data/rawdata/. The same yeast dataset was also used in example 2 to identify the target genes of transcription factors. In example 3, the circadian rhythm genes of *Arabidopsis thaliana* were identified using GEsture. Columbia diurnal gene expression data of *Arabidopsis thaliana* (Mockler Lab) measured in the condition of growing with 12h-light 12h-dark/24h-hot (COL_LDHH) was chosen (*Mockler et al., 2007*).

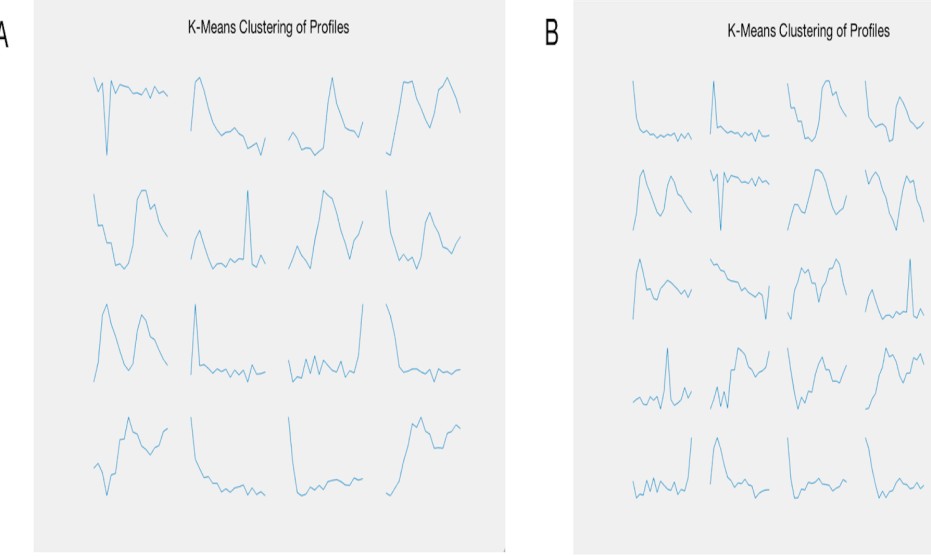

**Figure 5** **The resulting gene expression patterns of K-means clustering.** (A) represents 16 categories of clustering. (B) represents 25 categories of clustering.

## RESULTS

### Example 1: searching anticipated expression genes

Here, we used two methods, a user-defined pattern searching in GEsture and a K-means clustering method, to identify genes whose expression levels increased over time in the cell cycle. $K$-means clustering was first applied to cluster different gene expression categories. As shown in Fig. 5, a variety of gene expression patterns were identified at $k = 16$ and 25. However, the pattern of interest did not present itself in the results.

GEsture was then applied to find the genes showing the increasing pattern over time. We drew the anticipated expression curve in GEsture (Fig. 6A) and eleven genes were detected to express increasingly over time (Fig. 6B). Among these genes, four genes (YOR010C, YDR534C, YOR382W, and YNL066W) are the cellular component: cell wall proteins. Three genes are transporters (YHL047C, YMR058W, and YBR102C) (*Chervitz et al., 1999*), which may provide some hints for biologists to study biological processes of these genes and the transcriptional mechanisms of cell cycle regulation. In summary, this example demonstrated that GEsture was more straightforward and efficient for identifying genes that biologists are interested in, such as expression patterns whose expression levels increase with time during the cell cycle.

### Example 2: identifying target genes of transcriptional factors

As shown in the Saccharomyces Genome Database (SGD https://www.yeastgenome.org/), YNL309W(STB1) encodes a protein that contributes to the regulation of SBF and MBF target genes (*Chervitz et al., 1999*). During the G1/S transition in the cell cycle of yeast, SBF and MBF play the role of sequence-specific transcription factors in activating the gene expression (*Iyer et al., 2001*). We hypothesized that the genes showing similar, contrasting,

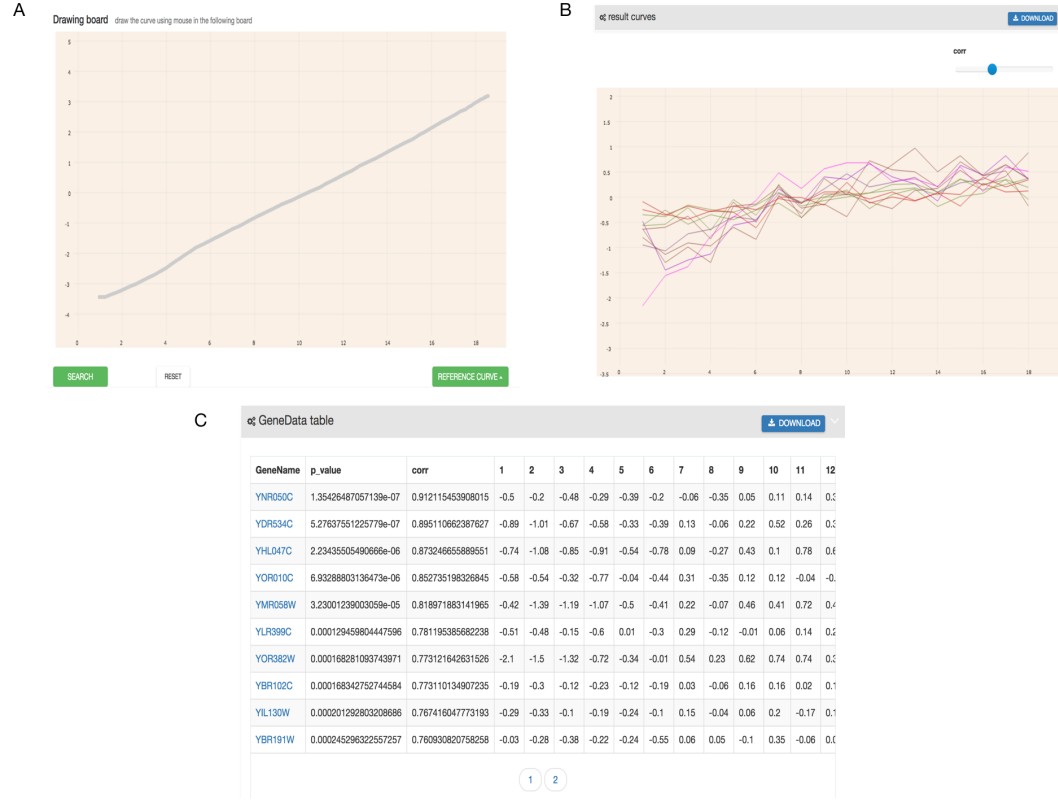

**Figure 6** **The hand-drawn curve and the results GEsture searched.** (A) represents the hand-drawn curve. (B) represents the result curves GEsture searched. (C) represents the result gene names GEsture searched.

or time-delayed expression pattern may be controlled by a similar regulatory mechanism. In this example, we searched genes using GEsture and explored whether there were other genes regulated by the same transcription factor.

We have drawn a curve like the expression of *YNL309W* (shown in Fig. 7) with the mouse and searched GEsture using three different patterns. Then we found 155 co-expression genes, 15 contrast expression genes and 44 one-interval shift expression genes. More detail information about these genes can be found in Table S1. YNL309W was detected in the co-expression gene list, which shows the accuracy of the tool.

We then used a database, YEASTRACT (Yeast Search for Transcriptional Regulators and Consensus Tracking http://www.yeastract.com/) (*Teixeira et al., 2006*), to assess which identified genes were regulated by the same TFs as YNL309W. YEASTRACT provides the known TF-target genes association of yeast in the cell-cycle process. *TEC1p* and *STE12p* were known TFs of *YNL309W* in the cell cycle (*Madhani et al., 1999*), TEC1p is responsible for positive regulation, and STE12p is responsible for negative regulation. After comparing the result, we found that 124 similar expression genes, five contrast expression genes and 27 shift expression genes (one interval) of the results above were regulated by TEC1p and STE12p (Table 1). Detailed gene information is listed in Table S2. The example indicated
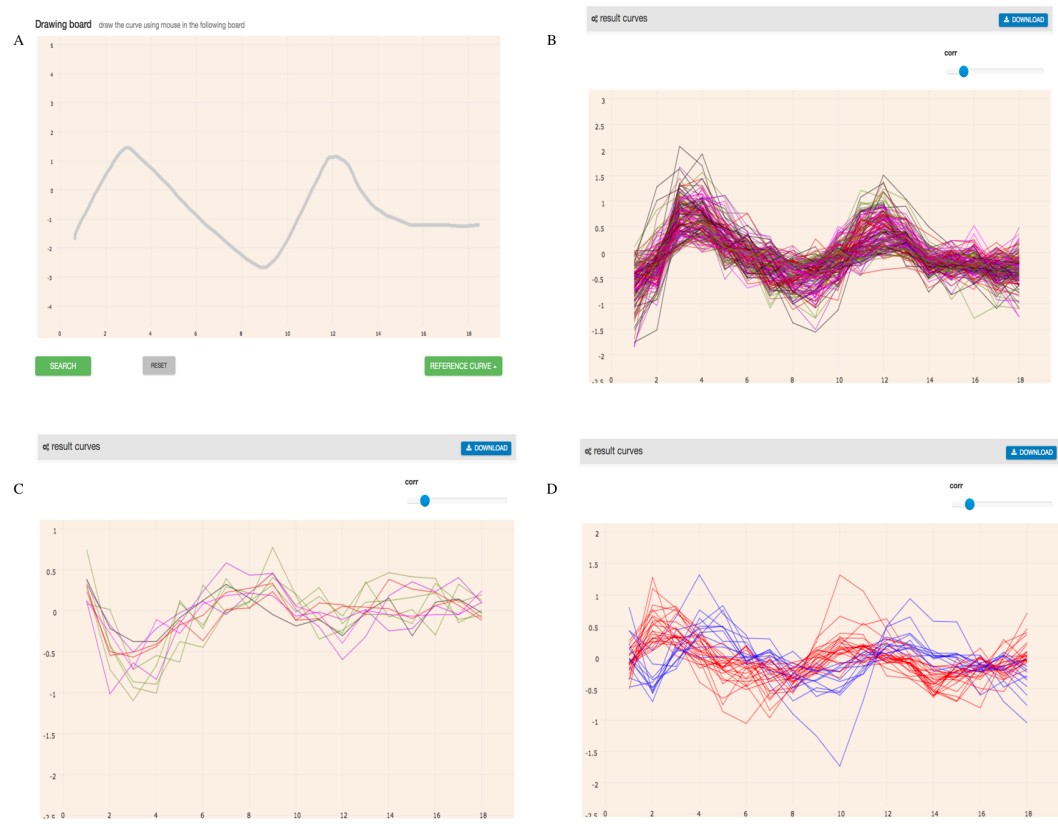

**Figure 7 Basic search results of three patterns of the curve like YNL309W.** In the coordinate system, the abscissa represents the count of sample and the ordinate represents gene expression value. (A) represents the expression curve like YNL309W. (B) represents the results of co-expression genes. (C) represents the results of opposite expression genes. (D) represents one interval shift expression genes. Blue line represents one time-delay interval co-expression genes and red line represents one time-ahead interval co-expression genes.

that similar expression genes may be regulated by the same transcription factors and GEsture can efficiently identify target genes associated with related transcription factors.

## Example 3: identifying circadian rhythm genes of *Arabidopsis thaliana*

In higher plants, the circadian rhythm phenomenon is a universal, intrinsic and autonomous timing mechanism of approximately 24-hours. This mechanism allows organisms to adapt to daily external changes in the environment, such as light, temperature and so on (*Bass & Takahashi, 2010*; *Bellpedersen et al., 2005*; *Hardin & Panda, 2013*; *Joska, Zaman & Belden, 2014*). The most noticeable characteristic of circadian rhythms is that the period of rhythm is close to 24 h in the absence of environmental stimuli. The expression pattern of circadian rhythms genes in the period of rhythm almost does not vary (*Hsu & Harmer, 2014*; *Wijnen & Young, 2006*). As of now, some circadian rhythms-associated genes of the *Arabidopsis thaliana* have been identified and cataloged by *The Arabidopsis Information Resource* (TAIR https://www.arabidopsis.org) (*Swarbreck et al., 2008*). Here,
**Table 1** **The ratio of target genes of TEC1p and STE12p in the total genes.** In the column Search Pattern, the numbers in the bracket represent the count of genes searched by GEsture which were recognized by YEASTRACT. In the column of numbers of genes, the number represents the count of target genes we searched out which were regulated by TFs. For example, 109 genes in co-expression genes were regulated by TEC1p and it accounted for 0.879 of total co-expression genes.

| Method | Search pattern | TF | Count | Ratio (%) |
|---|---|---|---|---|
| Draw the curve like YNL309W | Co-expression (124) | TEC1p | 109 | 87.9 |
| | | STE12p | 91 | 73.4 |
| | | TEC1p, STE12p | 76 | 61.3 |
| | Contrast expression (5) | TEC1p | 3 | 60.0 |
| | | STE12p | 5 | 100 |
| | | TEC1p, STE12p | 3 | 60.0 |
| | One interval shift expression (27) | TEC1p | 25 | 92.6 |
| | | STE12p | 22 | 81.5 |
| | | TEC1p, STE12p | 20 | 74.1 |

**Table 2** **Circadian rhythm genes searched by 3 patterns.**

| Search pattern | Count | Genes |
|---|---|---|
| Similar expression | 14 | AT5G02840, AT5G25830, AT4G24500, AT5G67380, AT5G15850, AT3G57040, AT5G05660, AT2G18915, AT3G50000, AT1G80820, AT5G56860, AT2G47700, AT3G55960, AT3G56480 |
| Contrast expression | 11 | AT3G06500, AT3G46640, AT2G42540, AT4G02630, AT3G52180, AT1G68050, AT4G26700, AT2G21660, AT3G46780, AT5G61380, AT2G18170 |
| Shift expression | 15 | AT4G24470, AT4G25100, AT4G30350, AT3G22170, AT1G10470, AT5G02120, AT1G27450, AT5G37260, AT1G15950, AT5G59560, AT3G07650, AT4G09970, AT2G25930, AT4G34680, AT3G61070 |

we input the expression pattern of known circadian rhythms genes and checked whether GEsture could efficiently identify genes related to circadian rhythm. As shown in Fig. 8, we drew an expression curve approximating circadian rhythm gene expression patterns. Three pattern searches were attempted for gene identification. The TAIR database was finally used to check whether the resulting genes from the pattern searches were related to circadian rhythm.

GEsture found 40 circadian rhythm genes using three search patterns (Table 2). Detailed information was listed in the Table S3. In these 40 circadian rhythm genes 14, 11, and 15 genes showing similar, contrast and shift patterns with the input mouse-entered pattern. Among the co-expression circadian rhythm genes, we have found *AT5G25830, AT5G15850, AT5G56860,* which are TFs of *Arabidopsis.* We also found some genes with an expression pattern similar to circadian rhythm but are not recognized as rhythm genes, such as *AT2G31990, AT1G32630, AT1G05320* and so on. The expression curves of these genes are similar to circadian rhythm genes, but their biological process is still shown as annotated in the TAIR database. The results may provide some hints for biologists to study biological functions and processes of these genes.

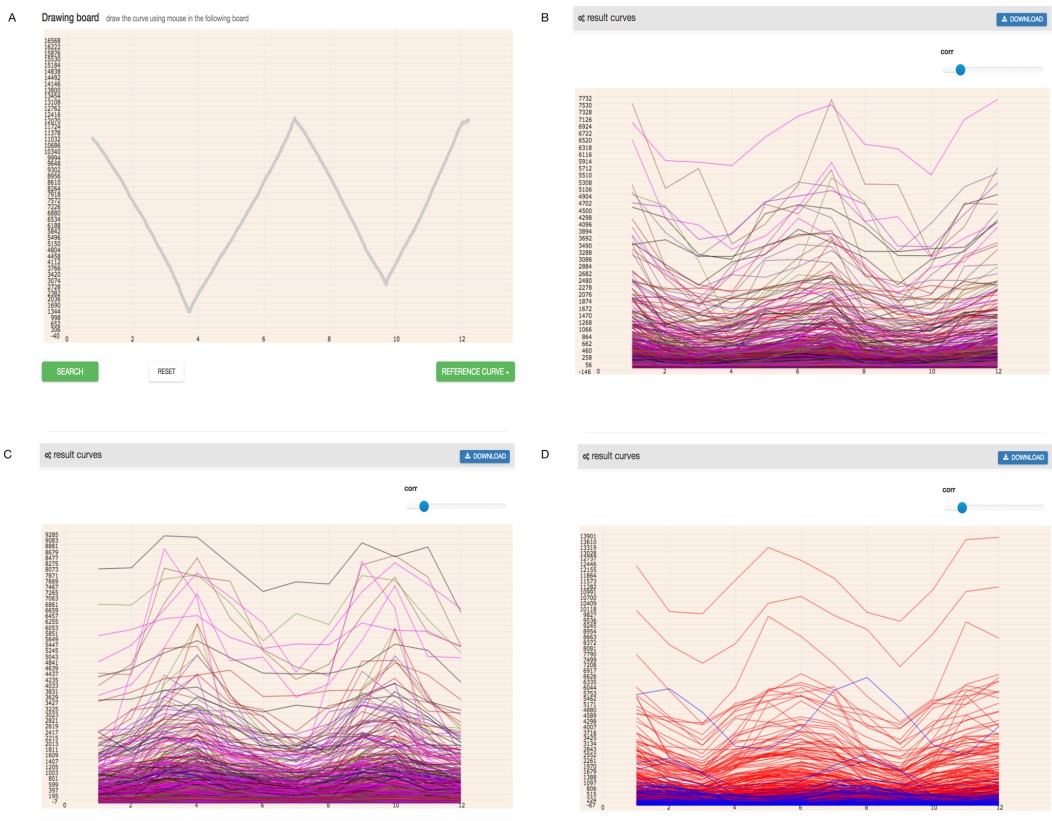

**Figure 8 Search circadian rhythm genes using three patterns.** In the coordinate system, the abscissa has 12 points representing the time point of every 4 h of 48 h, and the ordinate represents gene expression value. (A) a drawing curve which is acquired by imitating the expression of circadian rhythm genes (B) the result of similar expression genes, (C) the opposite expression genes (D) the shift expression genes.

## DISCUSSION

Two datasets, one from a yeast dataset, and the other from Arabidopsis thaliana were selected to assess the performance of our program. Three examples demonstrated the effectiveness of GEsture in searching co-expression, contrast and time-delay expression genes. The biological meaning of the output genes was explored. For reference, the two datasets, which are applied in the three examples are provided on the GEsture website.

GEsture is built for searching specific gene expression pattern from time-series gene expression data. The program was written in PHP, JAVASCRIPT, HTML5, and Bootstrap. Also, two plugins of cytoscape.js (*Franz et al., 2016*) and Echart.js were utilized for graphical visualization. Here, we used three examples to show that GEsture can search anticipated expression genes, target genes of transcriptional factors and circadian rhythm genes of *Arabidopsis thaliana*. The results of the first example indicate that clustering algorithms cannot efficiently dig out all gene expression patterns, some of which are hard to be clustered. Perhaps it is possible to identify the pattern by increasing the cluster number. However, it may require more time to attempt different cluster numbers and the process

is not efficient. While GEsture was shown more straightforward and efficient to identify genes by the way of drawing gene expression curves and it would be a good supplement tool of other clustering methods.

In the second example, GEsture searched similar expression genes by drawing a familiar gene expression curve rather than one concrete gene, such as an annotated gene name. It showed that GEsture was effective and efficient in exploring other genes with the similar expression patterns. Furthermore, about 73% ($(124+5+27)/(155+15+44)$) of overall result genes GEsture searched were controlled by the same TFs. The third example showed another function of GEsture, that not only was it capable of seeking target genes of the TFs, but it also performed well in detecting genes with similar functions by curves.

In short, GEsture provides an interactive interface for pattern searching and is convenient and easy for users to edit the gene expression curve, then further explore the similar expression genes matching the drawn expression curve. In contrast to inputting abstract parameters and data, it provides a visualization searching method to detect target genes and visualizes the result in heat map and network map furtherly. GEsture enriches the diversity of methods analyzing time-series expression data. It is available at http://bio.njfu.edu.cn/GEsture.

## CONCLUSIONS

In conclusion, GEsture is a web-based and user-friendly tool, which can detect gene expression patterns from time series gene expression data. It has some advantages over conventional analysis. First, users can quickly identify genes showing three expression patterns (similar, opposite, and shift) using a gene expression curve. Three examples showed that GEsture performed well. It can detect some expression patterns more efficiently than K-mean clustering. Therefore, GEsture will be an alternative method for users if the clustering methods failed. Second, GEsture provides an easy-to-use input interface. Users can draw a curve using a mouse instead of inputting abstract parameters from defined algorithms. Lastly, GEsture provides visualization tools (such as expression pattern figure, heatmap and correlation network) to display the searching results. The output results may provide useful information for users to understand the targets, function and biological processes of the input gene of choice.

## ACKNOWLEDGEMENTS

We thank the reviewers for their constructive advices for our paper content and software design. And we also thank Anna Jiang for her advice for designing this tool at the beginning. The data of Examples 1 and 2 was collected at the Yeast Cell Cycle Analysis Project. The data of Example 3 was collected from Mockler Lab.

### Funding

This study was supported by the National Key Research and Development Plan of China (2016YFD0600101), the 2017 Graduate Research and Innovation Program Projects in Jiangsu Province (KYCY17_0827) and the PAPD (Priority Academic Program Development) program at Nanjing Forestry University, Jiangsu Provincial Department of Housing and Urban-Rural Development (2016ZD44), the Fundamental Research Funds for the Central Non-Profit Research Institution of CAF (CAFYBB2017SZ001) and the National Natural Science Foundation of China (31570662, 31500533, and 61401214). The funders had no role in study design, data collection and analysis, decision to publish, or preparation of the manuscript.

### Grant Disclosures

The following grant information was disclosed by the authors:
National Key Research and Development Plan of China: 2016YFD0600101.
Graduate Research and Innovation Program Projects in Jiangsu Province: KYCY17_0827.
Nanjing Forestry University, Jiangsu Provincial Department of Housing and Urban-Rural Development: 2016ZD44.
Fundamental Research Funds for the Central Non-Profit Research Institution of CAF: CAFYBB2017SZ001.
National Natural Science Foundation of China: 31570662, 31500533, 61401214.

### Competing Interests

The authors declare there are no competing interests.

### Author Contributions

- Chunyan Wang and Yiqing Xu conceived and designed the experiments, performed the experiments, analyzed the data, contributed reagents/materials/analysis tools, prepared figures and/or tables, authored or reviewed drafts of the paper, approved the final draft.
- Xuelin Wang conceived and designed the experiments, performed the experiments, contributed reagents/materials/analysis tools, prepared figures and/or tables, authored or reviewed drafts of the paper, approved the final draft.
- Li Zhang and Suyun Wei conceived and designed the experiments, contributed reagents/materials/analysis tools, prepared figures and/or tables, authored or reviewed drafts of the paper, approved the final draft.
- Qiaolin Ye, Youxiang Zhu, Manoj Nainwal and Tongming Yin conceived and designed the experiments, contributed reagents/materials/analysis tools, authored or reviewed drafts of the paper, approved the final draft.
- Hengfu Yin conceived and designed the experiments, analyzed the data, contributed reagents/materials/analysis tools, authored or reviewed drafts of the paper, approved the final draft.

- Luis Tanon-Reyes and Feng Cheng conceived and designed the experiments, analyzed the data, contributed reagents/materials/analysis tools, authored or reviewed drafts of the paper, approved the final draft.
- Ning Ye conceived and designed the experiments, analyzed the data, contributed reagents/materials/analysis tools, prepared figures and/or tables, authored or reviewed drafts of the paper, approved the final draft.

### Data Availability

All the code of the website can be viewed at GitHub: https://github.com/15720613282/GEsture, and the raw data of examples can be found at:

https://github.com/15720613282/GEsture/tree/master/uploads/WCY. The files are also available at Figshare: Wang, Chunyan (2018): GEsture code.zip. figshare. Code. https://doi.org/10.6084/m9.figshare.6387839.v2.

### Supplemental Information

Supplemental information for this article can be found online at http://dx.doi.org/10.7717/peerj.4927#supplemental-information.

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
