# Peer review of "GEsture: an online hand-drawing tool for gene expression pattern search"

_PeerJ, doi:10.7717/peerj.4927_

## Round 0.1 · original submission · Major Revisions

In looking over this manuscript I found it a bit difficult to follow due to some of the grammar usage. I was able to make some changes which may clear up what I perceived to be the intended statements. I have attached an edited version of your manuscript with suggested changes highlighted. I did not necessarily review the abstract, but only tried to make it easier for potential reviewers to read. Please take a look at the manuscript version with suggested revisions and prepare an edited version. Once edited, please re-read it to make sure the message presented is in agreement with yours and re-submit the manuscript so that I may move it forward for review. Thank you for your understanding.

---

## Round 0.2 · Minor Revisions

Apologies for the delayed response on your manuscript. It seems many potential reviewers were very busy during this time of year; however, the following reviews were able to test the software in addition to reading the manuscript. I think it would be very important to seriously consider some of the suggestions, or at least, discuss restrictions of the software where the user may want to exceed its settings. The utility of this tool as a preview to big data sets does appear to be valuable and as searched unique in its application. Please consider the suggestions presented and prepare a revised manuscript. I will list this as requiring minor revisions; however, it may require a more extensive revisions to improve readability, and address some of the reviewer suggestions. Please try to refine the final edits to help shorten the turn-around time. Thank you for this contribution; it should be well received.

·

Basic reporting

The manuscript describes a web tool called GEsture that allows users to find gene expression patterns in time series data by hand-drawing the desired pattern. I see a clear need for such a tool to help biologists mine gene expression data, Being able to search for genes that show a contrasting expression pattern, or a time delayed pattern is also very nice. I could not find other tools that offer similar functionality through a web interface.

I had some difficulty running the tool with other data than the example files. Every time the uploading seemed to work fine, but upon clustering using K-means I got an undescriptive message saying: “sorry,please try again!”. I found out I that had to remove the header lines (which is mentioned in the manual) and take care of missing values in the data. Since the tool is meant for biologists without a bioinformatics background, the file upload functionality should be made more robust and get more helpful error messages.

Figure 4 is not very informative, why is this depicted as a network?

The English still needs quite some work, and I noticed two quoted sentences from published papers, which is rather unusual. Please find below my line by line comments and language corrections/suggestions (not exhaustive):
line 28: unclear what is meant with: “conforming to users’ expectations”
line 45: “genes expressing similarly expression” -> genes showing similar expression
line 52: “a CLARITY algorithm” -> the CLARITY algorithm
lines 56/57: unclear: “some genes that are high expression while the other genes are low expression at the same time”
line 57: Xia designed an eLSA package -> the eLSA package
line 59: “the clustering algorithm has” -> clustering algorithms have
lines 62-64: incomplete sentence followed by a literal quote, I would suggest to remove the quote and rewrite the sentence.
line 64/65: to my knowledge, clustering algorithms in general do not discard clusters with a small number of genes, please include a reference if the authors disagree.
line 67: unclear: “However, it cannot always cluster a category that is expected. Because a time-delayed phenomenon often appears in gene expression, K-means clustering cannot recognize it and
mistakenly divides it into many categories instead of classifying only one category.”
line 73: program searched -> program searches
line 78: user-friendly interface -> a user-friendly interface
line 93: with mouse in the drawing board -> with the mouse on the drawing board
line 98: pattern -> patterns
line 98: negative regulated genes -> negatively regulated genes
line 99: be founded -> be found
line 110: by user -> by the user
lines 161-163: unclear, should be rewritten: “This example demonstrated that GEsture was more straightforward and efficient to identify genes that were expected.”
line 169-170: rewrite: “‘SBF and MBF are sequence-specific transcription factors that activate gene expression during the G1/S transition of the cell cycle in yeast’, as suggested by Iyer et al.(2001) says.”
line 171: “the similar regulatory mechanism” -> “a similar regulatory mechanism”
line 207: information was listed -> information is listed
line 209: we have found AT5G25830, AT5G15850, AT5G56860 are -> which are
line 213: still annotated as in the -> still shown as annotated in the
line 219: “the three example data sets are provided on the Gesture website”: I could only find two?
line 222: “Programming was written in” -> “The program was written in”
lines 225-231: unclear
Legend figure 1: “List the the main operation procedures of this tool. Rectangular boxes labelled same color” -> “Lists the main operation procedures of this tool. Rectangular boxes with the same color
Figure 5: “The result gene” -> “The resulting gene”
Figure 7: Hand-drawing curve: hand-drawn curve. Not sure that “searched out.” is the proper expression here…
Table 1: the STE12p row should be 100% instead of 1.00%

Experimental design

Nothing is mentioned in the paper about any transformation of the data: should the uploaded data be log transformed (intensities/counts/FPKMs) or not? In figure 6 the data look log transformed, figure 7 the data seem normalised (and transformed), in figure 8 the data were untransformed and not normalised. Since log transformation is non-linear, this will affect the expected expression patterns.

Validity of the findings

Example 1 shows that the tool works, but the authors do not make clear why the 11 found genes “were expected” (line 163).

For Example 2 the authors use the yeast transcription factors TEC1p and STE12p to show that GEsture is able to find co-regulated genes, by claiming that 73% of the found genes are regulated by these two TFs. I am not sure that I use the YEASTRACT tool correctly, but I find in total 4900+ genes potentially regulated by the mentioned TFs. Assuming a total number of genes for yeast of 6600, this comes down to 74% of all genes, a percentage very close to the 73% the authors find. I would like to see more details for this example, like how many genes in total did the authors find to be regulated by TEC1p and STE12p, and for instance an enrichment test (i.e. Fisher’s exact test).

Additional comments

Web tool error messages in the web interface are not always clear/descriptive: when trying to upload a file with another extension than .csv, there is a “File Upload Error”, without specifying what the error is.

The icons on the csv files for removing and uploading do not seem to work (chrome/firefox on Mac).

For the graphs showing the expression patterns, it would be nice to have an option to show normalised curves (z-scores) in case the data were not normalised

·

Basic reporting

The writing is clear and easy to understand. The related studies were also sufficiently discussed and referenced. The figures and tables are also very appropriate and support authors' arguments.

Experimental design

no comment

Validity of the findings

no comment

Additional comments

The authors presented a web-server named GEsture, which can be used to query the expression
patterns from given input expression file. The users can either hand-draw
the expression patterns or choose the patterns from clustering results to get
all genes associated with those patterns. This tool is very user-friendly and functionally versatile.
I have only a few relatively minor concerns/comments and I hope the authors find them useful when revising the manuscript.

1. Yes, the clustering is very important to identify genes with similar expression.
In this work, the authors used K-means to cluster the genes based on the expression.
For the K-means, it is always a problem for users to specify K (# of clusters) for clustering.
It would be great if the GEsture can provide a suggestion of best K for clustering if users have no prior knowledge. Although this is a very hard problem and no perfect solution, still there are some works can be done here. For this task, maybe the authors can use some clustering metrics such as Silhouette or BIC to determine the best K for given input expression file.

2. The hand-drawn pattern searching functions are very useful.
But it seems that the expression is restricted to the range [-4,5].
What if users want to explore the patterns with much higher expression (say 8).
It would be great if the users can customize the expression range.

3. similar to the shift pattern search.
It would be great if the users are allowed to customize the shift range for searching.

4. Figure 4, can't read the text inside the node.
Please make it larger or change the color to make it more contrast to the background


5. Figure 8, The left size text is not readable, please enlarge the text

6. In table 1, how did the ratio is calculated?
It seems that there is something wrong
STE12P, 5 only accounts for 1% while TEC1p 3 accounts for 60%. shouldn't 1.00 here be 100%?

---

## Round 0.3 · accepted · Accept

Thank you for addressing the suggested revisions, the manuscript is now in a much easier-to-follow format. I feel that the manuscript is ready to move forward for publication as the tool should be made available for its helpful utility for time-series research. Please consider this manuscript accepted for publication. I did include some minor edits in the appended PDF which may help readability of the manuscript; although, I was somewhat concerned with the quality of the language in some of the sentences. With the suggested edits applied the general message should translate well to the user community. You can incorporate these edits while in production

Congratulations on your efforts; I feel this software will help interpretations of complex time-series experiments.

# #